# MODEL INVERSION ATTACKS ON VISION-LANGUAGE MODELS: DO THEY LEAK WHAT THEY LEARN?

## ABSTRACT

Model inversion (MI) attacks pose significant privacy risks by reconstructing private training data from trained neural networks. While prior works have focused on conventional unimodal DNNs, the vulnerability of vision-language models (VLMs) remains underexplored. In this paper, we conduct the first study to understand VLMs' vulnerability in leaking private visual training data. To tailored for VLMs' token-based generative nature, we introduce four novel token-based and sequence-based model inversion strategies. Particularly, we propose *Sequence-based Model Inversion with Adaptive Token Weighting (SMI-AW)*, based on our insight that not all tokens are equally informative for inversion. By dynamically reweighting token-level feedback according to each token's informativeness for inversion, SMI-AW achieves consistent improvement in reconstruction quality. Through extensive experiments and user study on three state-of-the-art VLMs and multiple datasets, we demonstrate, for the first time, that VLMs are susceptible to training data leakage. The experiments show that our proposed sequence-based methods, particularly SMI-AW combined with a logit-maximization loss based on vocabulary representation, can achieve competitive reconstruction and outperform token-based methods in attack accuracy and visual similarity. Importantly, human evaluation of the reconstructed images yields an attack accuracy of 75.31%, underscoring the severity of model inversion threats in VLMs. Notably, we also demonstrate inversion attacks on the publicly released VLMs. Our study reveals the privacy vulnerability of VLMs as they become increasingly popular across many applications such as healthcare and finance. **Our code, pretrained models, and reconstructed images are available in OpenReview's discussion forum.**

## 1 INTRODUCTION

Model Inversion (MI) attacks aim to reconstruct training data by exploiting information encoded within a trained model. These attacks pose significant privacy risks to unimodal DNNs (Fredrikson et al., 2015; Zhang et al., 2020; Chen et al., 2021; An et al., 2022; Struppek et al., 2022; Kahla et al., 2022; Han et al., 2023; Nguyen et al., 2023b; Yuan et al., 2023; Nguyen et al., 2023a; Qiu et al., 2024), The goal of MI attack is to reconstruct private training images $x$ associated with a target label $y$. These methods typically pose inversion as an optimization problem that maximizes the likelihood of $y$ under the target model:

$$\max_w \log \mathbb{P}_{M_{DNN}}(y \mid G(w)) \tag{1}$$

Here, $M_{DNN}$ is a unimodal DNN trained on private data $\mathcal{D}_{priv}$, and $G$ represents a generative model (Goodfellow et al., 2014; Karras et al., 2019). The optimization is usually accomplished by performing $N$ inversion update steps to generate a reconstruction $x^* = G(w^*)$ that approximates the training sample in $\mathcal{D}_{priv}$ for a given label $y$.

**Research Gap.** With the rapid advancement and widespread deployment of Vision-Language Models (VLMs) across various applications (Liu et al., 2024; Team et al., 2024; Bai et al., 2025), an important and timely question arises: *Are VLMs similarly vulnerable to Model Inversion attacks as unimodal DNNs?* In this context, we define an MI attack as the task of reconstructing VLM's training images by leveraging its textual input and output. Addressing this question is crucial for understanding and mitigating potential privacy threats in multimodal learning systems.

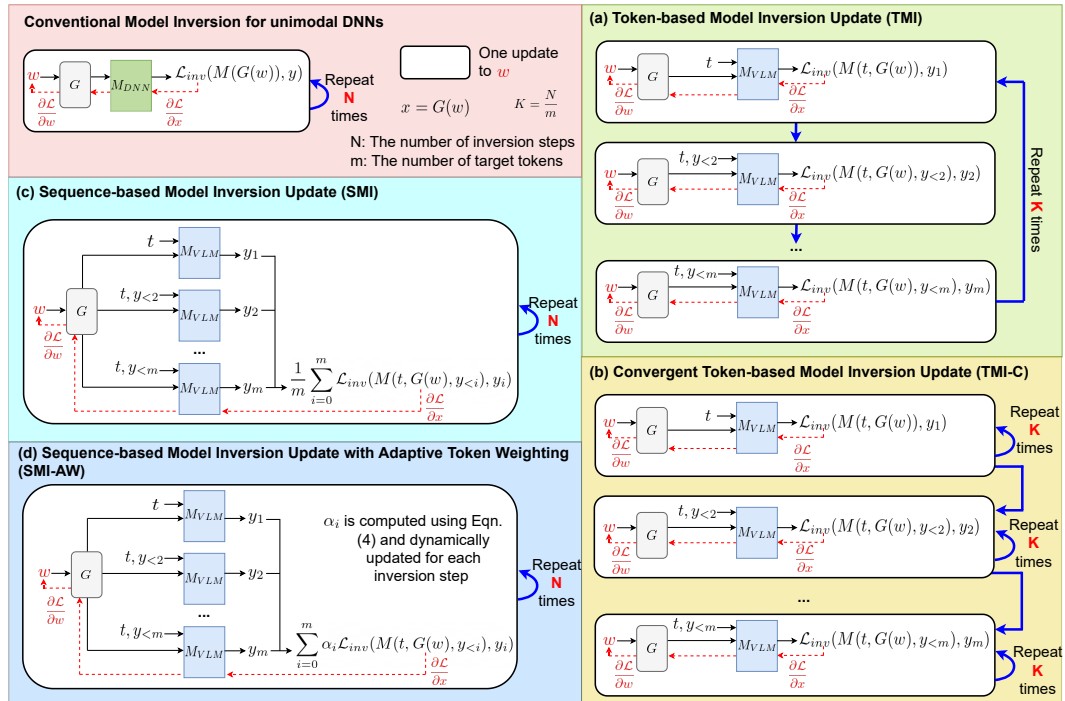

Figure 1: **Overview of our proposed Model Inversion attacks for VLMs.** Conventional MI typically targets unimodal DNNs, where the adversary seeks to reconstruct a training image $x = G(w)$ that maximizes the likelihood of a target class label $y$ under the target model $M_{DNN}$. The maximization is accomplished by repeating $N$ inversion steps to recover a high-fidelity reconstruction. In contrast, VLMs $M_{VLM}$ generate a sequence of tokens, and the target output $\mathbf{y} = (y_1, \ldots, y_m)$ is also a sequence of $m$ tokens. To address the unique nature of VLMs, we propose four MI strategies. **(a) Token-based Model Inversion (TMI):** We perform one gradient update to the latent variable $w$ after each generated token. This process continues for all $m$ tokens in the sequence, and the entire sequence-level inversion is repeated $K = N/m$ times. **(b) Convergent Token-based Model Inversion (TMI-C):** To ensure correctness of earlier tokens before generating subsequent ones, we propose updating $w$ for $K$ steps per token $y_i$, conditioning on the previous tokens $y_{<i}$. **(c) Sequence-based Model Inversion (SMI):** We compute one gradient update to $w$ based on the average loss over all $m$ tokens, providing a global view of the sequence-level gradients. **(d) Sequence-based Model Inversion with Adaptive Token Weighting (SMI-AW):** We introduce adaptive token weights $\alpha_i$ for each token $y_i$ to dynamically emphasize tokens that could provide more essential feedback signals, guiding reconstruction toward an image that matches the target description.

Unlike unimodal DNNs, vision-language models $M_{VLM}$ differ in several fundamental ways: they process multiple modalities (e.g., images and text), often comprise several distinct modules (e.g., separate encoders for vision and language, projector, language model), are often trained in multiple stages, and leverage broad, large-scale datasets. Crucially, a VLM's output is language, represented as a sequence of tokens. Consequently, MI attacks on VLMs must contend with unique aspects not present in unimodal DNNs. Furthermore, in unimodal DNNs, private visual features are directly embedded in the model parameters, increasing the risk that model inversion attacks can extract private visual features directly from the model. In contrast, many VLMs keep the vision encoder frozen during training and primarily update the language model. As a result, inversion attacks on VLMs rely on private information embedded in the language model's and projector's parameters to guide the image reconstruction, rather than directly extracting visual features from the vision encoder. These differences highlight a timely and important research gap: *the urgent need for novel Model Inversion tailored to the multimodal VLMs to understand their privacy threats*.

**In this work**, we introduce four novel token-based and sequence-based model inversion strategies tailored for VLMs (Figure 1). Our token-based attacks leverage token-level gradients to optimize the reconstructed images. In contrast, our sequence-based attacks utilize gradients aggregated over the entire sequence, offering a global perspective for image reconstruction. Particularly, we introduce

*Sequence-based Model Inversion with Adaptive Token Weighting (SMI-AW)*, which is based on our insight that not all tokens are equally informative for inversion. Low-confidence or mistaken tokens, rather than being noise, provide important feedback that highlights errors in the current reconstruction and effectively guide the search toward the correct image matching the target description. By dynamically reweighting token-level feedback at each inversion step to form the sequence-level feedback signal, SMI-AW achieves improved reconstruction quality.

We conduct experiments on three VLMs across three datasets to demonstrate the effectiveness of our inversion attacks. Notably, human evaluation of the reconstructed images achieves an attack accuracy of 75.31%, highlighting the severity of model inversion threats in VLMs. Furthermore, we validate the generalizability of our approach on publicly available VLMs, reinforcing its practical applicability and security implications. Our key contributions are as follows:

- We present a pioneering study of model inversion attacks on vision-language models, uncovering a novel security risk in the multimodal models.
- We introduce a suite of novel inversion strategies tailored for VLMs, including two Token-based MI (TMI and TMI-C) and two Sequence-based MI attacks (SMI and SMI-AW).
- The extensive experimental validation shows our proposed attacks, especially SMI-AW, achieve both high attack accuracy and good visual fidelity. Crucially, we showcase successful and high-fidelity inversion attacks against publicly available VLMs, underscoring the immediate and practical privacy risks posed by these models and the urgent need for robust defense mechanisms.

## 2 PROBLEM FORMULATION

We present the first comprehensive study of model inversion attacks in VLMs, which are increasingly used in real-world applications.

**Threat Model.** We consider a threat model where a VLM $M$ is pre-trained on broad data and fine-tuned on a private VQA dataset $\mathcal{D}_{priv} = \{(t, \mathbf{x}, y)\}$, where $\mathbf{x}$ is the image, $t$ and $y$ are the textual input and correct textual answer. For clarity, hereafter we use $M$ to denote a VLM and $M_{DNN}$ to refer to a unimodal DNNs. Using the tokenizer of $M$, the textual input $t$ and the textual answer $y$ are tokenized into sequences $\mathbf{t} = (t_1, t_2, \ldots, t_n)$ and $\mathbf{y} = (y_1, y_2, \ldots, y_m)$, respectively. We denote the full output sequence of $M$ given input $(\mathbf{t}, \mathbf{x})$ as $M(\mathbf{t}, \mathbf{x})$. The model's prediction of the $i$-th token $y_i$, conditioned on the previous tokens $y_{<i}$, is denoted by $M(\mathbf{t}, \mathbf{x}, y_{<i})$.

**Attacker's Goal.** Given a trained VLM $M$, the goal of a model inversion attack is to reconstruct a representative image $\mathbf{x}^*$ that reveals sensitive or private visual information from the private training image $\mathbf{x}$ in a data sample $(t, \mathbf{x}, y) \in \mathcal{D}_{priv}$. Specifically, the adversary is given access to the trained model $M$, a textual input prompt $t$, and the corresponding target output $y$. The target is to synthesize an image $\mathbf{x}^*$ such that $M(\mathbf{t}, \mathbf{x}^*) = \mathbf{y}$ where $\mathbf{t} = (t_1, t_2, \ldots, t_n)$ and $\mathbf{y} = (y_1, y_2, \ldots, y_m)$ are the output token sequence associated with the input textual $t$ and textual output $y$. In other words, the model inversion attack seeks to infer a plausible visual input $\mathbf{x}^*$ that, when paired with the given input tokens $\mathbf{t}$, produces the high likelihood output sequence $\mathbf{y}$. This reconstructed image $\mathbf{x}^*$ is intended to approximate or reveal private features of the true image $\mathbf{x}$, thereby compromising the visual confidentiality of the training data.

**Attacker's Capabilities.** We consider a white-box setting (Zhang et al., 2020; Chen et al., 2021; An et al., 2022; Struppek et al., 2022; Nguyen et al., 2023b; Qiu et al., 2024), where the attacker has full access to the VLM's architecture, parameters, output responses (e.g., generated text or logits), input prompts $t$, and their corresponding ground-truth answers $y$. The attacker also has access to an auxiliary public dataset $\mathcal{D}_{pub}$ from the same domain as $\mathcal{D}_{priv}$.

## 3 MODEL INVERSION STRATEGIES FOR VLMS

Given a VLM $M$ trained on broad data and fine-tuned with a private VQA dataset $\mathcal{D}_{priv} = \{(t, \mathbf{x}, y)\}$. Performing MI attacks directly in the image space is computationally expensive and often ineffective (Zhang et al., 2020). To reduce the search space of $x^*$, we follow conventional MI approaches for DNNs by leveraging a generative model $G$ trained on an auxiliary public dataset

**Algorithm 1 Token-based MI (TMI)**

1: **INPUT:** $M, G, \mathbf{t}, \mathbf{y} = (y_1, \ldots, y_m), N, \beta$
2: **OUTPUT:** $G(w)$
3: $K = N/m$
4: **for** $k = 1$ to $K$ **do**
5:     **for** $i = 1$ to $m$ **do**
6:         $\mathcal{L} = \mathcal{L}_{inv}(M(\mathbf{t}, G(w), y_{<i}), y_i)$   (2)
7:         $w = w - \beta \frac{\partial \mathcal{L}}{\partial w}$

**Algorithm 2 Convergent Token-based MI (TMI-C)**

1: **INPUT:** $M, G, \mathbf{t}, \mathbf{y} = (y_1, \ldots, y_m), N, \beta$
2: **OUTPUT:** $G(w)$
3: $K = N/m$
4: **for** $i = 1$ to $m$ **do**
5:     **for** $k = 1$ to $K$ **do**
6:         Compute $\mathcal{L}$ using Eqn. (2).
7:         $w = w - \beta \frac{\partial \mathcal{L}}{\partial w}$

$\mathcal{D}_{pub}$ (Zhang et al., 2020; Chen et al., 2021; Struppek et al., 2022; Nguyen et al., 2023b; Qiu et al., 2024). This allows us to shift the optimization from the high-dimensional image space to the lower-dimensional latent space of $G$, i.e., $x = G(w)$, where $w$ is the intermediate latent vector.

In contrast to conventional MI attacks targeting classification models, where the objective is to reconstruct an input image $x$ that yields a specific class label, *VLMs generate token sequences, and the target output also represented as a sequence of tokens*. This requires a reformulation of the MI objective to account for token generation. Our goal is to reconstruct a representative image $\mathbf{x}^* = G(w^*)$ by optimizing the latent vector $w$ such that the generated image captures the semantic content of the private training image $\mathbf{x}$ that associates with description $y$.

In this section, we model introduce four inversion strategies tailored for VLMs. The first two (TMI and TMI-C) are token-based approaches that leverage token-level gradients to optimize the reconstructed images. In contrast, the remaining two are sequence-based methods (SMI and SMI-AW) that aggregate gradients over the entire output sequence, providing a global perspective for inversion.

### 3.1 TOKEN-BASED MODEL INVERSION (TMI)

A natural approach is to treat the inversion process as a sequential update over individual token predictions. Given a target token sequence $\mathbf{y}$, we iteratively update the latent code $w$ after each generated token (see Figure 1 (a)). The details are in Algorithm 1. $N$ is the number of inversion steps, $\beta$ is the update rate of MI, $y_{<i}$ denotes the previous tokens. $\mathcal{L}_{inv}$ presents the inversion loss, guiding the generative model $G$ to produce images that induce the token $y_i$. We will discuss the design of $\mathcal{L}_{inv}$ in the next section. The optimization is performed over multiple iterations, typically up to a update limit of $N$ inversion steps. At each iteration, each token contributes independently to the optimization process.

### 3.2 CONVERGENT TOKEN-BASED MODEL INVERSION (TMI-C)

TMI performs a single update per token per iteration. However, VLMs generate each token $y_i$ based on the preceding tokens $y_{<i}$. To better align with this generative dependency, we propose Convergent Token-based Model Inversion (TMI-C), which updates the latent vector $w$ multiple times for each target token before proceeding to the next. Specifically, for each token $y_i$, we perform $K$ updates to $w$, thereby encouraging convergence of the token-level inversion subproblem before advancing to $y_{i+1}$ (see Figure 1 (b)). The details are presented in Algorithm 2.

### 3.3 SEQUENCE-BASED MODEL INVERSION (SMI)

Token-based model inversion methods treat each token independently, optimizing the latent vector $w$ based on individual token-level losses. As the output of VLMs is a sequence of tokens, we propose Sequence-based Model Inversion (SMI), which performs a single gradient update to $w$ by averaging the loss across all $m$ tokens in the sequence (see Figure 1 (c)). By aggregating token-level

**Algorithm 3 Sequence-based MI (SMI)**

1: **INPUT:** $M, G, \mathbf{t}, \mathbf{y} = (y_1, \ldots, y_m), N, \beta$
2: **OUTPUT:** $G(w)$
3: **for** $k = 1$ to $N$ **do**
4:     $\mathcal{L} = \frac{1}{m} \sum_{i=1}^{m} \mathcal{L}_{inv}(M(\mathbf{t}, G(w), y_{<i}), y_i)$   (3)
5:     $w = w - \beta \frac{\partial \mathcal{L}}{\partial w}$

losses into a unified objective, SMI leverages the interdependencies among tokens and provides more coherent gradients that reflects the structure of the full sequence. This global view encourages the model to recover a latent representation that is consistent across the entire sequence, rather than optimizing for each token in isolation. The details are presented in Algorithm 3.

### 3.4 Sequence-based Model Inversion with Adaptive Token Weighting (SMI-AW)

SMI in Eqn. (3) assumes that all tokens contribute equally to the inversion objective. In practice, however, some tokens are confidently predicted early during inversion, while others remain low-confident and potentially mispredicted. Importantly, these mistaken tokens provide essential feedback signals that guide the search toward a correct reconstructed image matching the target description. When uniform averaging is applied across all tokens, these signals are diluted and dampened, weakening the inversion gradients and slowing convergence.

To address this, we propose an adaptive token weighting scheme that amplifies the loss contributions from low-confidence (mispredicted) tokens and suppresses those with high-confidence (see Figure 1 (d)). Specifically, we adaptively reweight the token-wise loss using confidence-aware weights $\alpha_i$. The weights $\alpha_i$ are computed based on the predicted probability $\mathbb{P}(y_i)$ of token $y_i$ under the current model output. We define a token as *low-confidence* if $\mathbb{P}(y_i) < p_{thres}$, where $p_{thres}$ is a confidence threshold. Let $n$ be the number of such low-confidence tokens. The weights are then assigned as:

$$\alpha_i = \begin{cases} \begin{cases} \frac{1}{n}, & \text{if } \mathbb{P}(y_i) < p_{thres}, \\ 0, & \text{if } \mathbb{P}(y_i) \geq p_{thres} \end{cases} & \text{if } n > 0, \\ \\ \frac{1}{m}, & \text{if } n = 0. \end{cases} \tag{4}$$

This scheme dynamically focuses optimization on low-confidence tokens, amplifying gradient signals where prediction errors are more prominent. If there are no low-confidence tokens ($n = 0$), we set $\alpha_i = 1/m$, allowing the model to update $w$ with equal contributions from all tokens. The method is presented in Algorithm 4. **See Supp Sec C for further justification of SMI-AW via visual attention efficiency.**

---

**Algorithm 4 Sequence-based MI with Adaptive Token Weighting (SMI-AW)**

1: **INPUT:** $M, G, \mathbf{t}, \mathbf{y} = (y_1, \ldots, y_m), N, \beta, p_{thres}$
2: **OUTPUT:** $G(w)$
3: **for** $k = 1$ to $N$ **do**
4:     $n =$ the number of low-confidence token in $\mathbf{y}^{pred}$.
5:     Compute $\alpha_i$ for each token $y_i$ using Eqn. (4)
6:     $\mathcal{L} = \sum_{i=1}^{m} \alpha_i \mathcal{L}_{inv}(M(\mathbf{t}, G(w), y_{<i}), y_i)$  (5)
7:     $w = w - \beta \frac{\partial \mathcal{L}}{\partial w}$

---

**Remark.** To tailored for VLMs' token-based generative nature, we propose 4 token-based and sequence-based that leverage token-level and sequence-level gradients for image reconstruction.

### 3.5 Inversion Loss Design for VLMs

In this section, we present the adaptation of the inversion loss from conventional unimodal MI to VLMs. Specifically, the inversion loss in traditional MI typically consists of two components: $\mathcal{L}_{inv} = \mathcal{L}_{id} + \mathcal{L}_{prior}$, where the identity loss $\mathcal{L}_{id}$ guides the generator $G(w)$ to produce images that induce the label $y$ from the target model $M_{DNN}$, and $\mathcal{L}_{prior}$ is a regularization or prior loss. To extend this to VLMs, we focus on adapting the identity loss $\mathcal{L}_{id}$. We categorize it into two main types: cross-entropy-based and logit-based losses.

**Cross-entropy-based.** This loss is widely used in MI attacks (Zhang et al., 2020; Chen et al., 2021; Qiu et al., 2024) to optimize $w$ such that the reconstruction has the highest likelihood for the target class under the model $M$. For VLMs, we adapt the cross-entropy loss $\mathcal{L}_{CE}$ for each target token $y_i$ as follows:

$$\mathcal{L}_{CE}(M(\mathbf{t}, G(w), y_{<i}), y_i) = -\log \mathbb{P}_M(y_i | \mathbf{t}, G(w), y_{<i}) \tag{6}$$

$\mathbb{P}_M(y_i | \mathbf{t}, G(w), y_{<i})$ denotes the predicted probability of token $y_i$, computed over the tokenizer vocabulary of the VLM (e.g., LLaVa-v1.6 uses a vocabulary of 32,000 tokens).

**Logit-based.** Prior work shows that using cross-entropy loss in MI can lead to gradient vanishing (Yuan et al., 2023) or sub-optimal results (Nguyen et al., 2023b). To address this, Yuan et al. (2023) and Nguyen et al. (2023b) propose optimizing losses directly over logits of a target class. We adopt two such logit-based losses for VLMs: the Max-Margin Loss $\mathcal{L}_{MML}$ (Yuan et al., 2023) and the Logit-Maximization Loss $\mathcal{L}_{LOM}$ (Nguyen et al., 2023b) for a target token $y_i$:

$$\mathcal{L}_{MML}(M(\mathbf{t}, G(w), y_{<i}), y_i) = -l_{y_i}(\mathbf{t}, G(w), y_{<i}) + \max_{k \neq y_i} l_k(\mathbf{t}, G(w), y_{<i}) \tag{7}$$

$$\mathcal{L}_{LOM}(M(\mathbf{t}, G(w), y_{<i}), y_i) = -l_{y_i}(\mathbf{t}, G(w), y_{<i}) + \lambda \|f_{y_i} - f_{reg}\|_2^2 \tag{8}$$

Here, $l_{y_i}$ is the logit corresponding to the target token $y_i$, $\lambda$ is a hyperparameter, $f_{y_i} = M^{pen}(\mathbf{t}, G(w), y_{<i})$ where $M^{pen}()$ denotes the function that extracts the penultimate layer representations for a given input, and $f_{reg}$ is a sample activation from the penultimate layer $M^{pen}()$ computed using public images from $\mathcal{D}_{pub}$. Following (Nguyen et al., 2023b), the distribution of $f_{reg}$ is estimated over 2000 input pairs $(\mathbf{t}, \mathbf{x}_{pub})$, where $\mathbf{x}_{pub} \in \mathcal{D}_{pub}$. $\mathcal{L}_{MML}$ maximizes the logit of the correct token $y_i$ while penalizing the highest incorrect logit to mitigate gradient vanishing. On the other hand, $\mathcal{L}_{LOM}$ also maximizes the correct token's logit to avoid sub-optimality, while additionally penalizing deviations in the penultimate activations to prevent unbounded logits problem.

## 4 EXPERIMENTS

In this section, we evaluate the effectiveness of our 4 proposed model inversion attacks on 3 VLMs (i.e., LLaVA-v1.6, Qwen2.5-VL and MiniGPT-v2), 3 private datasets, 2 public datasets with an extensive evaluation spanning 5 metrics including the human evaluation.

### 4.1 EXPERIMENTAL SETTING

**Dataset.** Following standard model inversion (MI) setups(Zhang et al., 2020; Chen et al., 2021; Struppek et al., 2022; An et al., 2022; Nguyen et al., 2023b; Yuan et al., 2023; Struppek et al., 2024; Qiu et al., 2024; Ho et al., 2024; Koh et al., 2024), we use facial and fine-grained classification datasets to evaluate our approach. Specifically, we conduct experiments on three datasets: FaceScrub (Ng & Winkler, 2014), CelebA (Liu et al., 2015), and Stanford Dogs (Dataset, 2011). The FaceScrub dataset contains 106,836 images across 530 identities. For CelebA, we select the top 1,000 identities with the most samples from the full set of 10,177 identities. Stanford Dogs comprises images from 120 dog breeds, serving as a representative fine-grained visual dataset.

To train the target VLMs, we construct VQA-style datasets including VQA-FaceScrub, VQA-CelebA, and VQA-Stanford Dogs. For the facial datasets, each image $x$ is paired with a prompt $t =$ *"Who is the person in the image?"*, and the expected textual response $y$ is the individual's name (e.g., $y =$ "Candace Cameron Bure"). Since the CelebA dataset does not contain identity names, we randomly generate 1,000 unique English names, each comprising a distinct first and last name with no repetitions, and assign one to each identity in the selected CelebA subset. For VQA-Stanford Dogs, each image $\mathbf{x}$ is paired with a prompt $t =$ *"What breed is this dog?"*, and the target answer $y$ corresponds to the ground-truth breed label (e.g., "black-and-tan coonhound").

**Public Dataset and Image Generator.** For facial image reconstruction, we use FFHQ (Karras et al., 2019) as the public dataset $\mathcal{D}_{pub}$ and a pre-trained StyleGAN2 (Karras et al., 2020) trained on FFHQ. Following conventional MI (Struppek et al., 2022), we optimize in the latent space $w$ of StyleGAN2 to recover images $x = G(w)$. For Stanford Dogs experiments, we adopt AFHQ-Dogs (Choi et al., 2020) as $\mathcal{D}_{pub}$ to train the dog image generator.

**VLMs.** We fine-tune LLaVA-v1.6-7B (Liu et al., 2024), Qwen2.5VL-7B (Bai et al., 2025), and MiniGPT-v2 (Chen et al., 2023) using VQA-Facescrub, VQA-CelebA, and VQA-StanfordDogs.

**Evaluation Metrics.** To assess the quality of the inversion results, we adopt five metrics:

- **Attack accuracy.** We compute the attack accuracy using three frameworks as described below. We strictly follow the evaluation frameworks in their original works ( detailed setups in the Supp). Higher accuracy indicates a more effective inversion attack.

- **Attack accuracy evaluated by conventional evaluation framework** $\mathcal{F}_{DNN}$ ($AttAcc_D$ ↑) (Zhang et al., 2020; Chen et al., 2021; Struppek et al., 2022; Nguyen et al., 2023b; Qiu et al., 2024). This is a conventional framework, where the evaluation models are standard DNNs trained on private dataset. Following (Struppek et al., 2022; 2024), we use InceptionNet-v3 (Szegedy et al., 2016) as the evaluation model to classify reconstructed images, and compute the $Top1$ and $Top5$ based on whether the predicted label match the target label.
- **Attack accuracy evaluated by MLLM-based evaluation framework** $\mathcal{F}_{MLLM}$ ($AttAcc_M$ ↑). (Ho et al., 2025) demonstrate that $\mathcal{F}_{MLLM}$ can achieve better alignment with human evaluation. Unlike the conventional framework $\mathcal{F}_{DNN}$, which relies on the classification predictions of standard DNNs trained on private datasets, this metric leverages powerful MLLMs to evaluate the success of MI-reconstructed by referencing the corresponding private images.
- **Attack accuracy evaluated by human** $\mathcal{F}_{Human}$ ($AttAcc_H$ ↑). Following existing studies (An et al., 2022; Nguyen et al., 2023b), we conduct the user study on Amazon Mechanical Turk. Participants are asked to evaluate the success of MI-reconstructed by referencing the corresponding private images (Details in the Supp).

- **Feature distance.** We compute the $l_2$ distance between the feature representations of the reconstructed and the private training images (Struppek et al., 2022). Lower values indicate higher similarity and better inversion quality.

  - $\delta_{eval}$. Features are extracted by the evaluation model in $\mathcal{F}_{DNN}$.
  - $\delta_{face}$. Features are extracted by a pre-trained FaceNet model (Schroff et al., 2015).

## 4.2 RESULTS

We report attack results on the FaceScrub dataset in Table 1, evaluating four MI strategies under three inversion losses using LLaVa-1.6-7B. The results show that sequence-based mode inversion methods consistently outperform token-level MI approaches across all evaluation metrics. Among them, SMI-AW, when combined with the $\mathcal{L}_{LOM}$, achieves the highest performance. This highlights the advantage of employing adaptive token-wise weights that are dynamically updated at each inversion step. Using this method, we achieve an attack accuracy of 59.25% under $\mathcal{F}_{MLLM}$ while other distance metrics such as $\delta face$ and $\delta_{eval}$ are the lowest (where lower is better).

Results on additional datasets, including CelebA and Stanford Dogs, are shown in Table 2 using the logit maximization loss. We achieve high attack success rates, with attack accuracies of 66.91% on CelebA and 77.40% on Stanford Dogs. These findings are consistent with results on the FaceScrub dataset, where SMI-AW consistently achieves the highest attack performance across all metrics.

We further evaluate our proposed method on Qwen2.5-VL-7B and MiniGPT-v2, using the FaceScrub dataset (see Table 3). The results reinforce the generalizability of our findings, demonstrating that VLMs are broadly vulnerable to model inversion attacks. These results underscore the severity of this vulnerability and raise a significant alarm about the susceptibility of VLMs to inversion-based privacy breaches.

## 4.3 ANALYSIS

To better understand why token-based MI methods underperform compared to sequence-based approaches, we analyze the match rate between the final reconstructed images $M(\mathbf{t}, G(w^*))$ and the corresponding target textual answers $y$. Specifically, we define the match rate as the percentage of reconstructed images for which the target answer $y$ appears as a substring of the predicted text associated with the image. In other words, it reflects the proportion of reconstructions whose generated text aligns with the target textual answer at the end of the inversion process.

The results, shown in Figure 2, reveal a clear distinction between the two types of methods. Token-based MIs exhibit poor convergence behavior, with match rates ranging from 60% to 79% for TMI, and dropping below 30% for TMI-C. In contrast, sequence-based methods such as SMI and SMI-AW achieve match rates exceeding 95%, indicating more reliable alignment between reconstructed images and their textual targets. It is important to note that a high match rate does not necessarily

imply a successful attack, as the optimization may overfit or converge to a poor local minimum. Nevertheless, a higher match rate generally correlates with a greater likelihood of a successful identity inversion attack.

Table 1: Comparison of performance metrics across four inversion strategies using LLaVa-1.6-7B fine-tuned on the FaceScrub dataset, evaluated with three identity losses. We highlight the best results in bold.

| $\mathcal{L}_{inv}$ | $AttAcc_M \uparrow$ | $AttAcc_D \uparrow$ | | $\delta_{face} \downarrow$ | $\delta_{eval} \downarrow$ |
|---|---|---|---|---|---|
| | | Top1 | Top5 | | |
| **TMI** | | | | | |
| $\mathcal{L}_{CE}$ | 37.78% | 17.71% | 39.79% | 0.8939 | 147.35 |
| $\mathcal{L}_{MML}$ | 39.98% | 17.31% | 38.51% | 0.9065 | 193.14 |
| $\mathcal{L}_{LOM}$ | 44.34% | 21.77% | 44.69% | 0.8488 | 141.87 |
| **TMI-C** | | | | | |
| $\mathcal{L}_{CE}$ | 21.77% | 6.39% | 18.58% | 1.0911 | 636.50 |
| $\mathcal{L}_{MML}$ | 25.99% | 6.51% | 18.82% | 1.0659 | 205.71 |
| $\mathcal{L}_{LOM}$ | 31.16% | 9.32% | 24.22% | 1.0221 | 457.49 |
| **SMI** | | | | | |
| $\mathcal{L}_{CE}$ | 40.97% | 18.25% | 41.11% | 0.8682 | 144.53 |
| $\mathcal{L}_{MML}$ | 55.52% | 32.83% | 60.12% | 0.7569 | 137.43 |
| $\mathcal{L}_{LOM}$ | 59.17% | 33.47% | 61.89% | 0.7465 | 140.83 |
| **SMI-AW** | | | | | |
| $\mathcal{L}_{CE}$ | 44.17% | 22.33% | 46.63% | 0.8464 | 145.29 |
| $\mathcal{L}_{MML}$ | 57.15% | 34.91% | 61.84% | 0.7444 | 138.24 |
| $\mathcal{L}_{LOM}$ | **59.25%** | **36.98%** | **64.69%** | **0.7286** | **135.90** |

Table 2: We report the results on the CelebA and Stanford Dogs dataset across four inversion strategies with $\mathcal{L}_{LOM}$.

| Method | $AttAcc_M \uparrow$ | $AttAcc_D \uparrow$ | | $\delta_{face} \downarrow$ | $\delta_{eval} \downarrow$ |
|---|---|---|---|---|---|
| | | Top1 | Top5 | | |
| **CelebA dataset** | | | | | |
| TMI | 39.74% | 15.31% | 33.14% | 1.0195 | 428.66 |
| TMI-C | 18.73% | 3.63% | 10.29% | 1.2370 | 446.90 |
| SMI | 64.93% | 38.30% | 63.69% | 0.8294 | 416.34 |
| SMI-AW | **66.91%** | **40.83%** | **65.84%** | **0.8133** | **415.25** |
| **Stanford Dogs dataset** | | | | | |
| TMI | 61.46% | 40.31% | 70.21% | - | 102.40 |
| TMI-C | 48.54% | 29.69% | 59.79% | - | 102.23 |
| SMI | 75.94% | 53.65% | 82.19% | - | **76.98** |
| SMI-AW | **77.40%** | **58.33%** | **86.04%** | - | 78.61 |

Table 3: We report the results of Qwen2.5-VL-7B and MiniGPT-v2 on the Facescub dataset. Here we use SMI-AW with $\mathcal{L}_{LOM}$.

| $M$ | $AttAcc_M \uparrow$ | $AttAcc_D \uparrow$ | | $\delta_{face} \downarrow$ | $\delta_{eval} \downarrow$ |
|---|---|---|---|---|---|
| | | Top1 | Top5 | | |
| MiniGPT | 50.80% | 15.26% | 34.69% | 0.8909 | 161.35 |
| Qwen2.5 | 36.42% | 14.91% | 31.37% | 1.0115 | 144.92 |

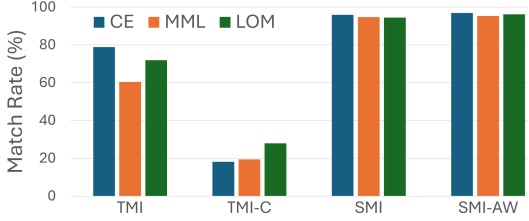

Figure 2: The match rate between the output text of the reconstructed image and the target output text $y$.

Table 4: Human evaluation results. We evaluate our SMI-AW method using $\mathcal{L}_{LOM}$, the private datasets $\mathcal{D}_{priv}$ are FaceScrub and CelebA.

| VLM | $\mathcal{D}_{priv}$ | $AttAcc_H \uparrow$ |
|---|---|---|
| LLaVA-v1.6-7B | | 75.31% |
| MiniGPT-v2 | Facescrub | 61.84% |
| Qwen2.5-VL | | 57.74% |
| LLaVA-v1.6-7B | CelebA | 61.95% |

## 4.4 QUALITATIVE RESULTS

Figure 3 shows qualitative results demonstrating the effectiveness of our method. Using SMI-AW with $\mathcal{L}_{LOM}$, the reconstructed images from the LLaVA-v1.6-7B model (second row) closely resemble the corresponding identities in $\mathcal{D}_{priv}$ (first row). This strong visual similarity highlights the ability of our model inversion approach to recover identifiable features from the training data. **More reconstructed images of other models and datasets can be found in Supp.**

## 4.5 HUMAN EVALUATION

We further conduct human evaluation on reconstructed images using two datasets Facescrub and CelebA. Each user study involves 4,240 participants for the FaceScrub dataset and 8,000 participants for the CelebA dataset. The results show that 57.74% to 75.31% of the reconstructed samples are deemed successful attacks, i.e., human annotators recognize the generated images as depicting the same identity as those in the private image set (see Table 4). This highlights the alarming potential of such inversion attacks to compromise sensitive identity information.

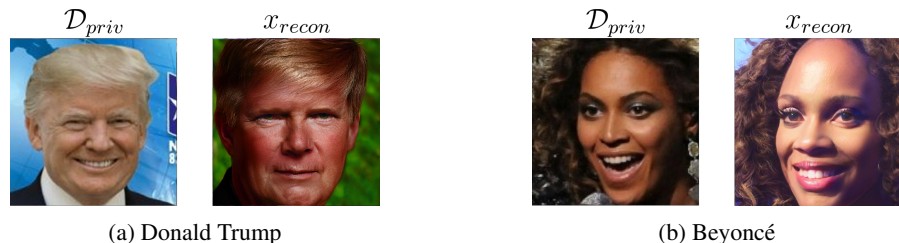

Figure 3: Qualitative results on the Facescrub dataset using the SMI-AW and $\mathcal{L}_{LOM}$. The first row shows images from the private training dataset, while the second row presents the reconstructed images corresponding to each individual in the first row. The visual similarity between the original and reconstructed images demonstrates the effectiveness of our inversion method in recovering private training data. **More reconstructed images can be found in Supp.**

### 4.6 EVALUATION WITH PUBLICLY RELEASED VLM

In the experiments above, we fine-tuned the target model using a private training dataset following prior MI work on conventional DNNs (Chen et al., 2021; Nguyen et al., 2023b; Struppek et al., 2022; Qiu et al., 2024). In this section, we extend our analysis to the publicly available LLaVA-v1.6-7B model, aiming to reconstruct potential training images directly from it.

Figure 4 shows the results of our best setup of MI attack, SMI-AW using the logit maximization loss. The target is to reconstruct images of some identities that appear in the training dataset of the LLaVA-v1.6-7B model. We present two image pairs: in each pair, the left image is a training sample of an identity, while the right image shows the corresponding reconstruction generated by the publicly available model. The visual similarity between the pairs indicates that the pre-trained VLM may reveal identifiable information from its training data, exposing a vulnerability to model inversion attacks. **More results can be found in Supp.**

|  $\mathcal{D}_{priv}$ | $x_{recon}$ | | $\mathcal{D}_{priv}$ | $x_{recon}$ |
|---|---|---|---|---|

(a) Donald Trump          (b) Beyoncé

Figure 4: We reconstruct images of (a) Donald Trump and (b) Beyoncé from the pre-trained LLaVA-v1.6-7B model. We use SMI-AW with $\mathcal{L}_{LOM}$ to reconstruct images. For each pair, the left image shows a training image of Donald Trump or Beyoncé, while the right image presents the reconstruction obtained via our model inversion attack. This result illustrates that the pre-trained VLM is vulnerable to training data leakage through model inversion. **More results can be found in Supp.**

## 5 CONCLUSION

This study pioneers the investigation of model inversion attacks on Vision-Language Models, demonstrating for the first time their susceptibility to leaking private visual training data. Our novel token-based and sequence-based inversion strategies reveal significant privacy risks across state-of-the-art and publicly available VLMs. Particularly, our proposed Sequence-based Model Inversion with Adaptive Token Weighting (SMI-AW) achieve an attack accuracy of 75.31%. These findings underscore the urgent need for robust privacy safeguards as VLMs become more prevalent in real-world applications. **Additional analysis, limitation and broader impact are included in Supp.**

## REPRODUCIBILITY STATEMENT

In accordance with ICLR policy, our code, pretrained models, and reconstructed images are made anonymously available for review in Openreview's discussion forums. To further ensure reproducibility, we will release the code and pretrained models publicly upon publication. Comprehensive details of our model architecture, experimental setup, and hyperparameters are included in the main paper and elaborated in Section A of the Supplementary Material.

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

# Supplementary material

In this supplementary material, we provide additional experiments, analysis, ablation study, and details that are required to reproduce our results. These are not included in the main paper due to space limitations.

CONTENTS

## A    RESEARCH REPRODUCIBILITY DETAILS

In accordance with ICLR policy, our code, pretrained models, and reconstructed images are made anonymously available for review in the discussion forums.

### A.1    HYPERPARAMETERS

To fine-tune the VLMs, we follow the standard hyperparameters provided in the official implementations of LLaVA-v1.6-Vicuna-7B[1] (Liu et al., 2024), Qwen2.5-VL-7B[2] (Bai et al., 2025), and MiniGPT-v2[3] (Chen et al., 2023). Fine-tuning is conducted on the VQA-FaceScrub, VQA-CelebA, and VQA-StanfordDogs datasets.

For the attacks, we use $N = 70$ inversion steps for experiments on the LLaVA-v1.6-7B model, and $N = 100$ for MiniGPT-v2 and Qwen2.5-VL-7B. The inversion update rate $\beta = 0.05$. We set the confidence threshold $p_{thres} = 0.999$ for all experiments using the logit maximization loss $\mathcal{L}_{LOM}$. Additional results with varying values of $p_{thres}$ are provided in the ablation study section (Supp).

To compute the regularization term $f_{reg}$ in Eqn. (8), we follow (Nguyen et al., 2023b) by using 2,000 images from a public dataset $\mathcal{D}_{pub}$ to estimate the mean and variance of the penultimate layer activations of the VLMs.

### A.2    COMPUTATIONAL RESOURCES

All experiments were conducted on NVIDIA RTX A6000 Ada GPUs running Ubuntu 20.04.2 LTS, equipped with AMD Ryzen Threadripper PRO 5975WX 32-core processors. The environment setup for each model is provided in the official implementations of the VLMs, including: LLaVA-v1.6-Vicuna-7B (Liu et al., 2024), Qwen2.5-VL-7B (Bai et al., 2025), and MiniGPT-v2 (Chen et al., 2023).

To evaluate $AttAcc_M$, we strictly follow the protocol in (Ho et al., 2025), using the Gemini 2.0 Flash API. In total, we evaluate 95,200 MI-reconstructed images for our main experiments (main paper).

## B    ADDITIONAL RESULTS

### B.1    EXTENDED EVALUATION ON PUBLICLY RELEASED VLM

In this section, we extend our analysis to the publicly available LLaVA-v1.6-7B model (Liu et al., 2024), aiming to reconstruct training images from accessing the model only.

Figure S.1 shows the results of our best setup of MI attack, SMI-AW using the logit maximization loss $\mathcal{L}_{LOM}$. The target is to reconstruct images of celebrities that appear in the training dataset of the LLaVA-v1.6-7B model. To reconstructed images from the model, we use the textual input $t =$ "Identify the person in the image and return only their name?" and the target textual answer is a celebrity's name, i.e $y =$ "Beyoncé".

We visualize image pairs: in each pair, the right image is the reconstruction generated from the publicly available model, and the left image is a training image of an individual. We emphasize that the training dataset is fully unknown and inaccessible for the inversion attack. The visual similarity between the pairs indicates that the pre-trained VLM may reveal identifiable information from its training data, exposing a vulnerability to model inversion attacks.

### B.2    ADDITIONAL QUALITATIVE RESULTS

Reconstructed images from the FaceScrub dataset using three VLMs, LLaVA-v1.6-7B, MiniGPT-v2, and Qwen2.5-VL, are shown in Figure S.2, Figure S.3, and Figure S.4, respectively. For the

---

[1] https://github.com/haotian-liu/LLaVA

[2] https://github.com/QwenLM/Qwen2.5-VL

[3] https://github.com/Vision-CAIR/MiniGPT-4

$\mathcal{D}_{priv}$  $x_{recon}$

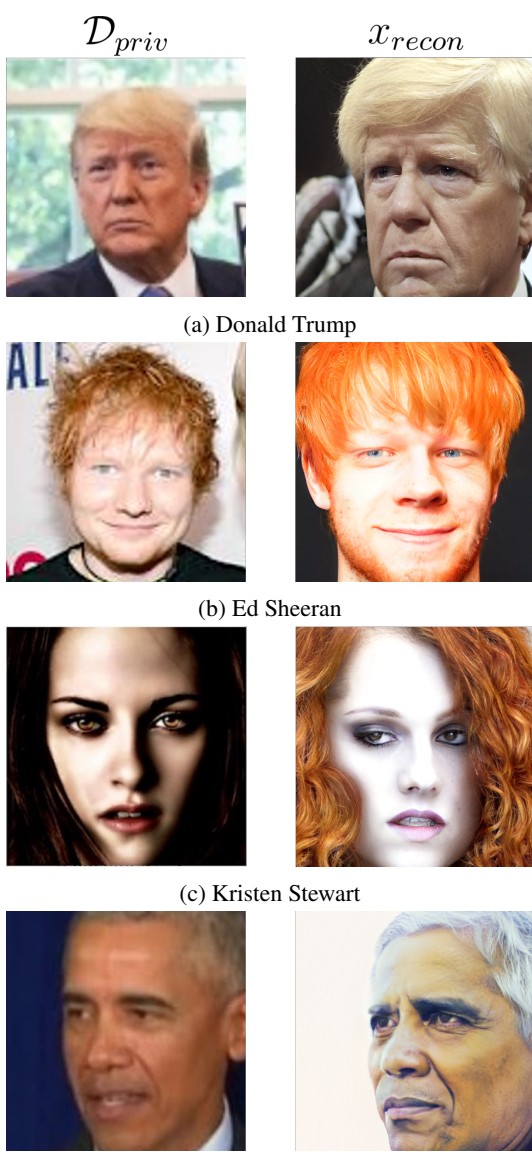

(a) Donald Trump

(b) Ed Sheeran

(c) Kristen Stewart

(d) Barack Obama

Figure S.1: Reconstructed images using our SMI-AW with $\mathcal{L}_{LOM}$ on the publicly available LLaVA-v1.6-7B model. Each pair consists of a reconstructed image (right) and a corresponding training image (left) in the training dataset of LLaVA-v1.6-7B model. We emphasize that the training dataset is fully unknown and inaccessible for the inversion attack. The strong similarity suggests the pre-trained VLM may leak identifiable training data, exposing it to model inversion attacks.

CelebA and Stanford Dogs datasets, reconstructed images using LLaVA-v1.6-7B are presented in Figure S.5 and Figure S.6. All reconstructions are generated using SMI-AW with the logit maximization loss $\mathcal{L}_{LOM}$.

For each pair, the left column shows images from the private training dataset, while the right column presents the reconstructed images corresponding to each individual in the left column. Qualitative results demonstrate the effectiveness of our method. This strong visual similarity highlights the ability of our model inversion approach to recover identifiable features from the training data.

## C  VISUAL ATTENTION EFFICIENCY ANALYSIS FOR SMI-AW

In this section, we further validate our SMI-AW via visual attention efficiency analysis.

Given a target description, the inversion process produces a sequence of output tokens. Some tokens are predicted with high confidence early on, while others remain low-confidence and are often mis-predicted. In this section, we demonstrate that these low-confidence tokens play a critical role: they provide essential feedback signals that guide the search toward a reconstructed image that better aligns with the target description. When uniform averaging is applied across all tokens, however, these informative signals are diluted, weakening the inversion gradients and slowing convergence.

### C.1  MEASURING VISUAL ATTENTION EFFICIENCY

During inversion, we collect the attention score distributions of each output token $y_i$ across all layers and aggregate the scores corresponding to image tokens (visual attentions) (Chen et al., 2024). We define the visual attention efficiency of an output token $y_i$ as:

$$\epsilon^i = \sum_{j=0}^{N} \frac{\alpha_{\text{img}}^{i,j}}{|\text{img}|}, \tag{9}$$

where $N$ is the number of layers, and $\alpha_{\text{img}}^{i,j}$ denotes the visual attention scores assigned to image tokens when predicting $y_i$ at layer $j$.

During inversion, the reconstructed image is iteratively refined through gradient feedback by aligning the predicted output tokens with the target tokens. The gradients propagate through the image tokens encoded by the vision encoder of the target VLM. Therefore, *an output token with a higher visual attention score indicates that the VLM regards the corresponding image tokens as more relevant for predicting $y_i$.* Such tokens can provide essential feedback signals that guide the inversion process toward reconstructing an image more faithfully aligned with the target description.

### C.2  LOW-CONFIDENCE TOKENS HAVE HIGHER VISUAL ATTENTION EFFICIENCY

Our objective is to analyze the *visual attention efficiency* of output tokens, i.e, how strongly the image tokens contribute to predicting each output token.

We categorize output tokens into **low-confidence** and **high-confidence** groups. For each token $y_i$, we compute its confidence with respect to the ground-truth token. A token is considered *low-confidence* if $\mathbb{P}(y_i) < p_{thres}$ and *high-confidence* otherwise. In addition, we measure the visual attention score $\epsilon^i$ for each token $y_i$. A score is classified as *inefficient* if it falls below the mean attention score across all $M$ output tokens, and as *efficient* otherwise.

We evaluate this analysis on 530 identities from the FaceScrub dataset, setting the confidence threshold to $p_{thres} = 0.999$. Figure S.7 compares the visual attention maps of low- versus high-confidence tokens. The results reveal that low-confidence tokens consistently exhibit stronger visual attention than high-confidence tokens. For a quantitative perspective, Table S.1 summarizes the relationship between attention efficiency and confidence. A clear pattern emerges: high-confidence tokens generally align with inefficient visual attention, while low-confidence tokens are more strongly associated with efficient visual attention.

This finding further validates our adaptive weighting strategy (SMI-AW). **Because low-confidence tokens correlate with efficient visual attention, they provide stronger gradient signals.** By assigning greater weight to low-confidence tokens, we guide the inversion process toward reconstructions that more faithfully capture the target description. In particular, we compare the percentage of efficient visual attention tokens contributing to the reconstruction of the inverted image between SMI and SMI-AW (see Figure S.8). In SMI, all tokens are used to update the image, resulting in only around 38% of them have efficient visual attention. By contrast, SMI-AW employs adaptive token weighting to focus more on tokens with efficient visual attention, ranging from 75% to 92% of the tokens used to optimize the inverted images, which provide stronger gradient signals for the inversion process.

Table S.1: We summarize the relationship between the predicted confidence of output tokens and the attention efficiency of image tokens. Our observations show that high-confidence tokens typically correspond to low attention efficiency, whereas low-confidence tokens tend to correspond to high attention efficiency.

|  | Low-confidence | High-confidence |
|---|---|---|
| Inefficient Visual Attention | 18.19 % | **49.06 %** |
| Efficient Visual Attention | **31.38 %** | 1.37 % |

## D  ABLATION STUDY

### D.1  ABLATION STUDY ON $p_{thres}$

We conduct an ablation study to investigate the effect of setting $p_{thres}$ in SMI-AW. Here, we use $M$ = LLaVA-v1.6-7B, $\mathcal{D}_{priv}$ = Facescrub, $\mathcal{L}_{inv} = \mathcal{L}_{LOM}$. As shown in Table S.2, using a higher threshold to focus on tokens with low confidence scores consistently improves attack performance across all evaluation metrics. For all experiments in main paper, we use $p = 0.999$.

Table S.2: Ablation study on $p_{thres}$ for adaptive token weights in SMI-AW. Here, we use $M$ = LLaVA-v1.6-7B, $\mathcal{D}_{priv}$ = Facescrub, $\mathcal{L}_{inv} = \mathcal{L}_{LOM}$. Using a higher threshold to focus on tokens with low confidence scores consistently improves attack performance across all evaluation metrics.

| $p_{thres}$ | $AttAcc_M \uparrow$ | $AttAcc_D \uparrow$ | | $\delta_{face} \downarrow$ | $\delta_{eval} \downarrow$ |
|---|---|---|---|---|---|
|  |  | $Top1$ | $Top5$ |  |  |
| 0.999 | **59.83%** | **37.17%** | **65.31%** | **0.7349** | **135.81** |
| 0.98 | 57.05% | 34.32% | 61.23% | 0.7486 | 137.27 |
| 0.95 | 56.96% | 33.21% | 61.86% | 0.7495 | 136.71 |

### D.2  ERROR BAR

We repeat each experiment three times using different random seeds and report the results in Table S.3. Specifically, we use $M$ = LLaVA-v1.6-7B, $\mathcal{D}_{priv}$ = Facescrub, and $p_{thres} = 0.999$. The results demonstrate that our attacks have low standard deviation.

Table S.3: Error bars for our two model inversion strategies SMI and SMI-AW. Each experiment was repeated 3 times, and we report the mean and standard deviation of the attack performance. Here, we use $M$ = LLaVa-v1.6-7B, $\mathcal{D}_{priv}$ = Facescrub, $p_{thres} = 0.999$. All inversion strategies are combined with logit maximization loss $\mathbf{L}_{LOM}$.

| Method | $AttAcc_M \uparrow$ | $AttAcc_D \uparrow$ | | $\delta_{face} \downarrow$ | $\delta_{eval} \downarrow$ |
|---|---|---|---|---|---|
|  |  | $Top1$ | $Top5$ |  |  |
| SMI | $57.83 \pm 1.18\%$ | $33.50 \pm 0.19\%$ | $61.56 \pm 0.30\%$ | $0.7473 \pm 0.0006$ | $137.89 \pm 2.62$ |
| SMI-AW | $59.39 \pm 0.39\%$ | $37.00 \pm 0.17\%$ | $65.01 \pm 0.31\%$ | $0.7318 \pm 0.0031$ | $135.84 \pm 0.05$ |

## E  EXPERIMENTAL SETTING

### E.1  EVALUATION METRICS

In this section, we provide a detailed implementation for five metrics used in our work to access MI attacks.

- **Attack accuracy.** Attack accuracy measures the success rates of MI attacks. Following existing literature, we compute attack accuracy via three frameworks:

- **Attack accuracy evaluated by conventional evaluation framework** $\mathcal{F}_{DNN}$ ($AttAcc_D$ ↑) (Zhang et al., 2020; Chen et al., 2021; Struppek et al., 2022; Nguyen et al., 2023b; Qiu et al., 2024). Following (Struppek et al., 2022; 2024), we use InceptionNet-v3 (Szegedy et al., 2016) as the evaluation model. For a fair comparison, we use the identical checkpoints of InceptionNet-v3 for Facescrubs, CelebA and Stanford Dogs from (Struppek et al., 2022) for evaluation of each dataset. We report *Top-1* and *Top-5* Accuracy.
  - **Attack accuracy evaluated by MLLM-based evaluation framework** $\mathcal{F}_{MLLM}$ ($AttAcc_M$ ↑). (Ho et al., 2025) demonstrate that $\mathcal{F}_{MLLM}$ can achieve better alignment with human evaluation than $\mathcal{F}_{DNN}$ ($AttAcc_D$ ↑ by mitigating Type-I adversarial transferability. The evaluation involves presenting a reconstructed image (image A) and a set of private reference images (set B) to an MLLM (e.g., Gemini 2.0 Flash), and prompting it with the question: "Does image A depict the same individual as images in set B?" If the model responds "Yes", the attack is considered successful. An example query is shown in Fig. S.9.
  - **Attack accuracy evaluated by human** $\mathcal{F}_{Human}(AttAcc_H$ ↑). Following existing studies (An et al., 2022; Nguyen et al., 2023b), we conduct the user study on Amazon Mechanical Turk. Participants are asked to evaluate the success of MI-reconstructed by referencing the corresponding private images. Similar to $\mathcal{F}_{MLLM}$, it involves presenting an image A and a set of images B. They are asked to answer "Yes" or "No" to indicate whether image A depicts the same identity as images in set B (see Fig. S.9). Each image pair is shown in a randomized order and displayed for up to 60 seconds. Each user study involves 4,240 participants for the FaceScrub dataset and 8,000 participants for the CelebA dataset.
- **Feature distance.** We compute the $l_2$ distance between the feature representations of the reconstructed and the private training images (Struppek et al., 2022). Lower values indicate higher similarity and better inversion quality.
  - $\delta_{eval}$. Features are extracted by the evaluation model as used in $\mathcal{F}_{DNN}$.
  - $\delta_{face}$. Features are extracted by a pre-trained FaceNet model (Schroff et al., 2015).

### E.2 INITIAL CANDIDATE SELECTION

Following the method from (Struppek et al., 2022), we perform an initial selection to identify promising candidates for inversion. We begin by sampling 2000 latent vectors, denoted as $\{w\}_{i=1}^{2}000$, from the prior distribution. For each $w$, we evaluate the target VLMs loss. We then select the top $n$ vectors with the lowest loss to serve as our initialization candidates. In our experiments, we set $n = 16$ to create 16 candidates for attacks.

### E.3 FINAL SELECTION

To select the final reconstructed image, we perform a final selection step, also following the method from (Struppek et al., 2022). This step aims to identify the reconstructed images that have the highest confidence. For each of the $n$ initialization candidates, we apply 10 random data augmentations and re-evaluate the target VLMs loss. We calculate the average loss for each candidate across these augmentations and select the $n/2$ candidates with the lowest average loss as the final attack outputs.

## F RELATED WORK

**Model Inversion.** Model Inversion (MI) seeks to recover information about a model's private training data via pretrained model. Given a target model $M$ trained on a private dataset $\mathcal{D}_{\text{priv}}$, the adversary aims to infer sensitive information about the data in $\mathcal{D}_{\text{priv}}$, despite it being inaccessible after training. MI attacks are commonly framed as the task of reconstructing an input $x_y^r$ that the model $M$ would classify as belonging to a particular label $y$. The foundational MI method is introduced in (Fredrikson et al., 2014), demonstrating that machine learning models could be exploited to recover patients' genomic and demographic data.

**Model Inversion in Unimodal Vision Models.** Model Inversion (MI) has been extensively studied to reconstruct private training images in unimodal vision models. For example, in the context of

face recognition, MI attacks attempt to recover facial images that the model would likely associate with a specific individual.

Building on the foundational work of (Fredrikson et al., 2014), early MI attacks targeting facial recognition are proposed in (Fredrikson et al., 2015; Yang et al., 2019), demonstrating the feasibility of reconstructing recognizable facial images from the outputs of pretrained models. However, performing direct optimization in the high-dimensional image space is challenging due to the large search space. To address this, recent advanced generative-based MI attacks have shifted the search to the latent space of deep generative models (Zhang et al., 2020; Wang et al., 2021a; Chen et al., 2021; Yang et al., 2019; Yuan et al., 2023; Nguyen et al., 2023b; Struppek et al., 2022; Qiu et al., 2024).

Specifically, GMI (Zhang et al., 2020) and PPA (Struppek et al., 2022) employ WGAN (Arjovsky et al., 2017) and StyleGAN (Karras et al., 2019), respectively, trained on an auxiliary public dataset $\mathcal{D}_{\text{pub}}$ that similar to the private dataset $\mathcal{D}_{\text{priv}}$. The pretrained GAN is served as prior knowledge for the inversion process. To improve this prior knowledge, KEDMI (Chen et al., 2021) trains inversion-specific GANs using knowledge extracted from the target model $M$. PLGMI (Yuan et al., 2023) introduces pseudo-labels to enhance conditional GAN training. IF-GMI (Qiu et al., 2024) utilizes intermediate feature representations from pretrained GAN blocks. Most recently, PPDG-MI (Peng et al., 2024) improves the generative prior by fine-tuning GANs on high-quality pseudo-private data, thereby increasing the likelihood of sampling reconstructions close to true private data. Beyond improving GAN-based priors, several studies focus on improving the MI objective including max-margin loss (Yuan et al., 2023) and logit loss (Nguyen et al., 2023b) to better guide the inversion process. Additionally, LOMMA (Nguyen et al., 2023b) introduces the concept of augmented models to improve the generalizability of MI attacks.

Unlike MI attacks, MI defenses aim to reduce the leakage of private training data while maintaining strong predictive performance. Several approaches have been proposed to defend against MI attacks. MID (Wang et al., 2021b) and BiDO (Peng et al., 2022) introduce regularization-based defenses that include the term of regularization in the training objective. The crucial drawback of these approaches is that the regularizers often conflict with the training objective resulting in a significant degradation in model's utility. Beyond regularization-based strategies, TL-DMI (Ho et al., 2024) leverages transfer learning to improve MI robustness, and LS (Struppek et al., 2024) applies Negative Label Smoothing to mitigate inversion risks. Architectural approaches to improve MI robustness have also been explored in (Koh et al., 2024). More recently, Trap-MID (Liu & Chen, 2024) introduces a novel defense by embedding trapdoor signals into $M$. These signals act as decoys that mislead MI attacks into reconstructing trapdoor triggers instead of actual private data.

**Model Inversion in Multimodal Large Vision-Language Models.** Large Vision-Language Models (VLMs) are increasingly deployed in many real-world applications across diverse domains, including sensitive areas. Unlike unimodal vision models, VLMs are designed to process both image and text inputs and generate text responses. A typical VLM architecture includes a text tokenizer to encode textual inputs into text tokens, a vision encoder to extract image features as image tokens, and a lightweight projection layer that maps image tokens into the text token space. These tokens are then concatenated and passed through a LLM to produce the final response. This multimodal processing pipeline fundamentally distinguishes VLMs from traditional unimodal vision models.

As VLMs are being adopted more widely, including in privacy-sensitive scenarios, understanding their potential vulnerability to data leakage via MI attacks becomes critical. **However, while MI attacks have been extensively studied in unimodal vision models, to the best of our knowledge, there has been no prior work investigating MI attacks on multimodal VLMs. To fill this gap,** *we conduct the first study on MI attacks targeting VLMs and propose a novel MI attack framework specifically tailored to the multimodal setting of VLMs.*

# G DISCUSSION

## G.1 BROADER IMPACTS

Our work reveals, for the first time, that VLMs are vulnerable to MI attacks. As VLMs are increasingly deployed in many applications including sensitive domains, this poses serious privacy risks.

Although our work focuses on developing a new MI attack for VLMs, we also provide a fundamental understanding for the development of MI defenses in multimodal systems. We hope this work encourages the community to incorporate privacy audits in VLM deployment and to pursue principled model design that mitigates data leakage.

Our methods are intended solely for research and defense development. We strongly discourage misuse and emphasize responsible disclosure when evaluating model vulnerabilities.

### G.2 LIMITATIONS

While following conventional MI attacks to focus on facial images and dog breeds, a more diverse domain scenarios, such as natural scenes or medical images, remain an important direction for future research. Moreover, evaluations on a broader range of models are needed to further comprehend our study on MI for VLMs.

## H THE USE OF LLMS

This manuscript was edited using LLMs for language polishing and writing improvements. The authors retain full responsibility for the research content, including the concepts, analyses, and conclusions.

$\mathcal{D}_{priv}$  $x_{recon}$      $\mathcal{D}_{priv}$  $x_{recon}$

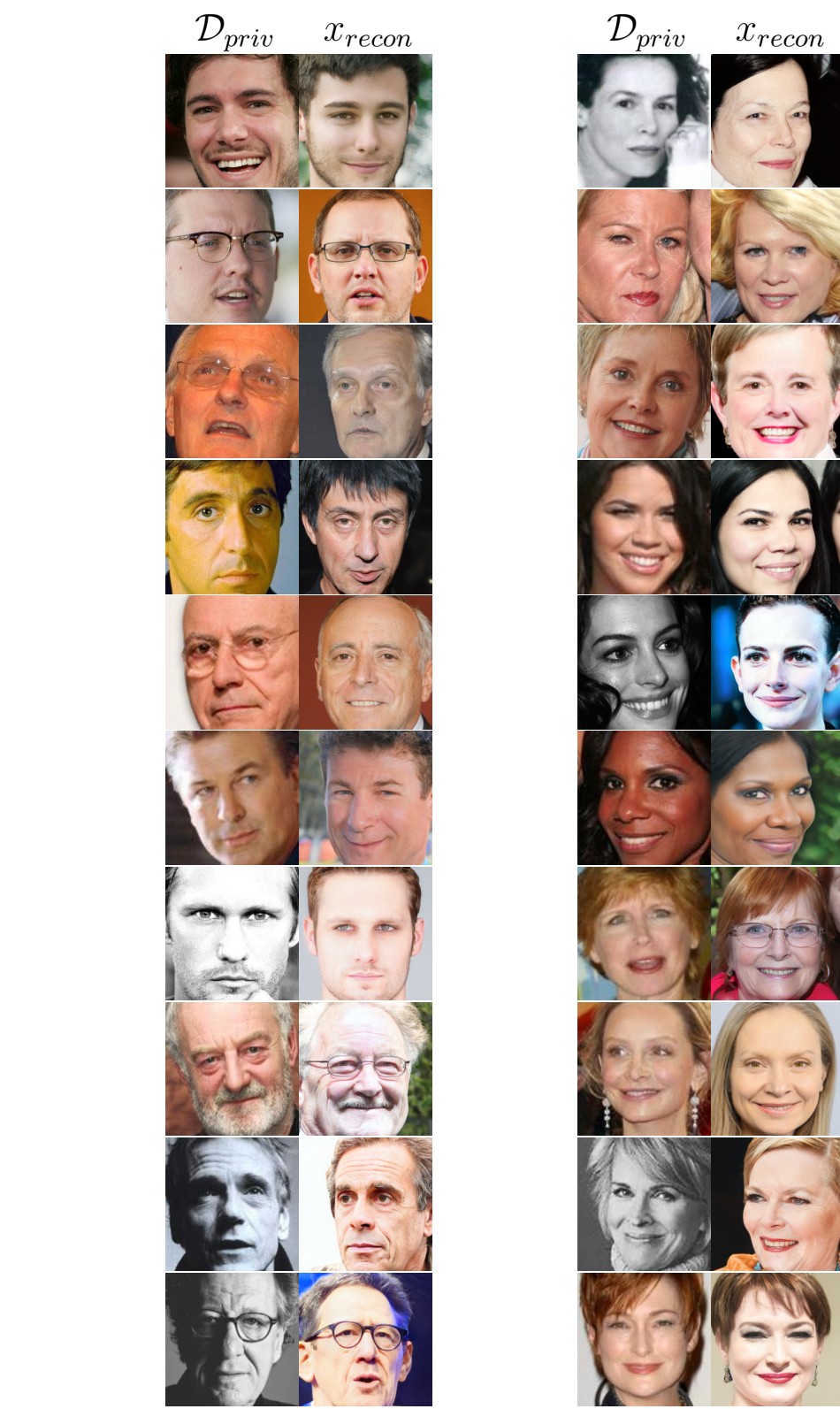

Figure S.2: Qualitative results on Facescrub dataset using the SMI-AW and $\mathcal{L}_{LOM}$, $M$ = LLaVA-v1.6-7B. For each pair, the left column shows images from the private training dataset, while the right column presents the reconstructed images corresponding to each individual in the left column.

$\mathcal{D}_{priv}$ $\quad x_{recon}$ $\qquad$ $\mathcal{D}_{priv}$ $\quad x_{recon}$

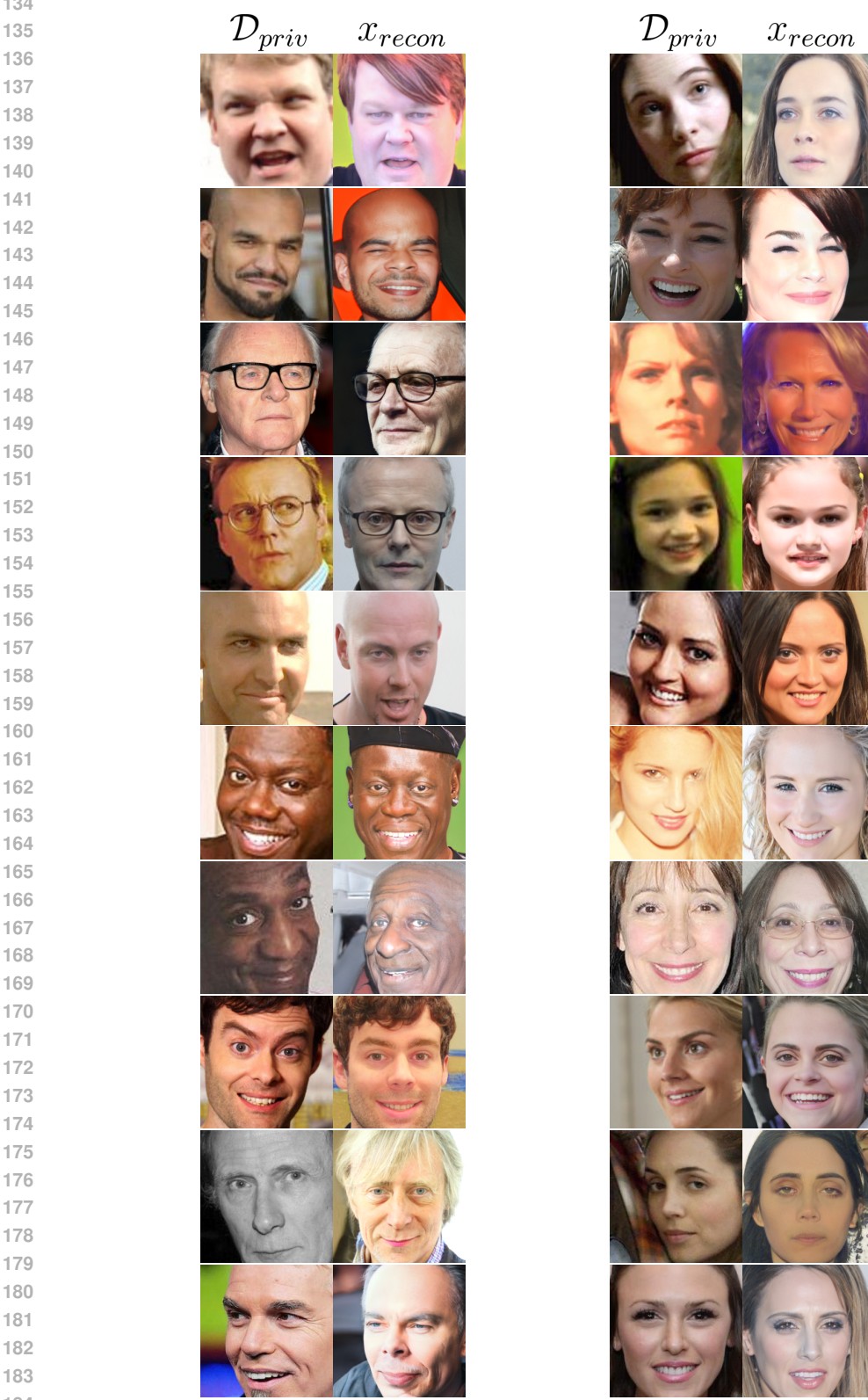

Figure S.3: Qualitative results on Facescrub dataset using the SMI-AW and $\mathcal{L}_{LOM}$, $M$ = MiniGPT-v2. For each pair, the left column shows images from the private training dataset, while the right column presents the reconstructed images corresponding to each individual in the left column.

$\mathcal{D}_{priv}$   $x_{recon}$     $\mathcal{D}_{priv}$   $x_{recon}$

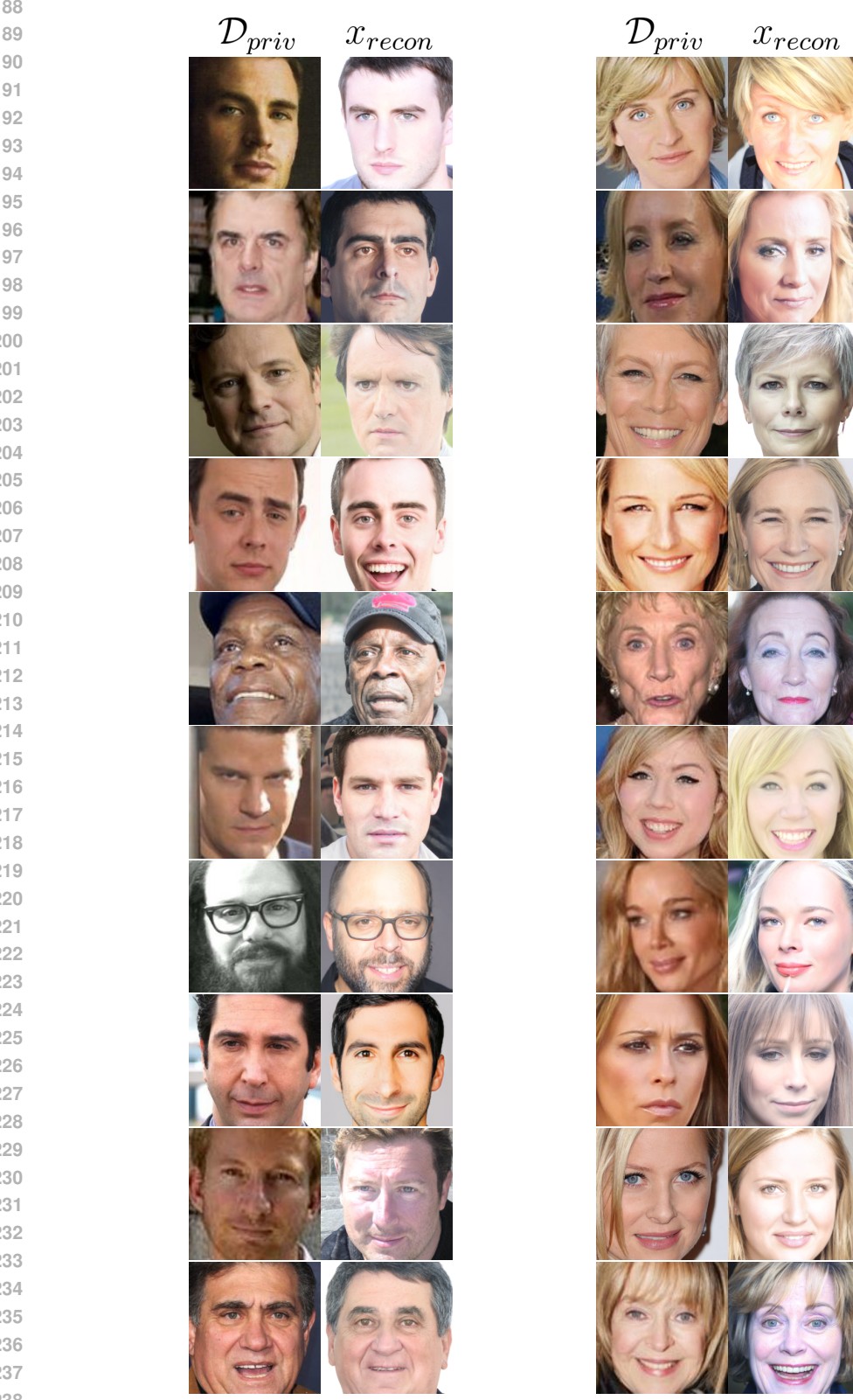

Figure S.4: Qualitative results on Facescrub dataset using the SMI-AW and $\mathcal{L}_{LOM}$, $M$ = Qwen2.5-VL. For each pair, the left column shows images from the private training dataset, while the right column presents the reconstructed images corresponding to each individual in the left column.

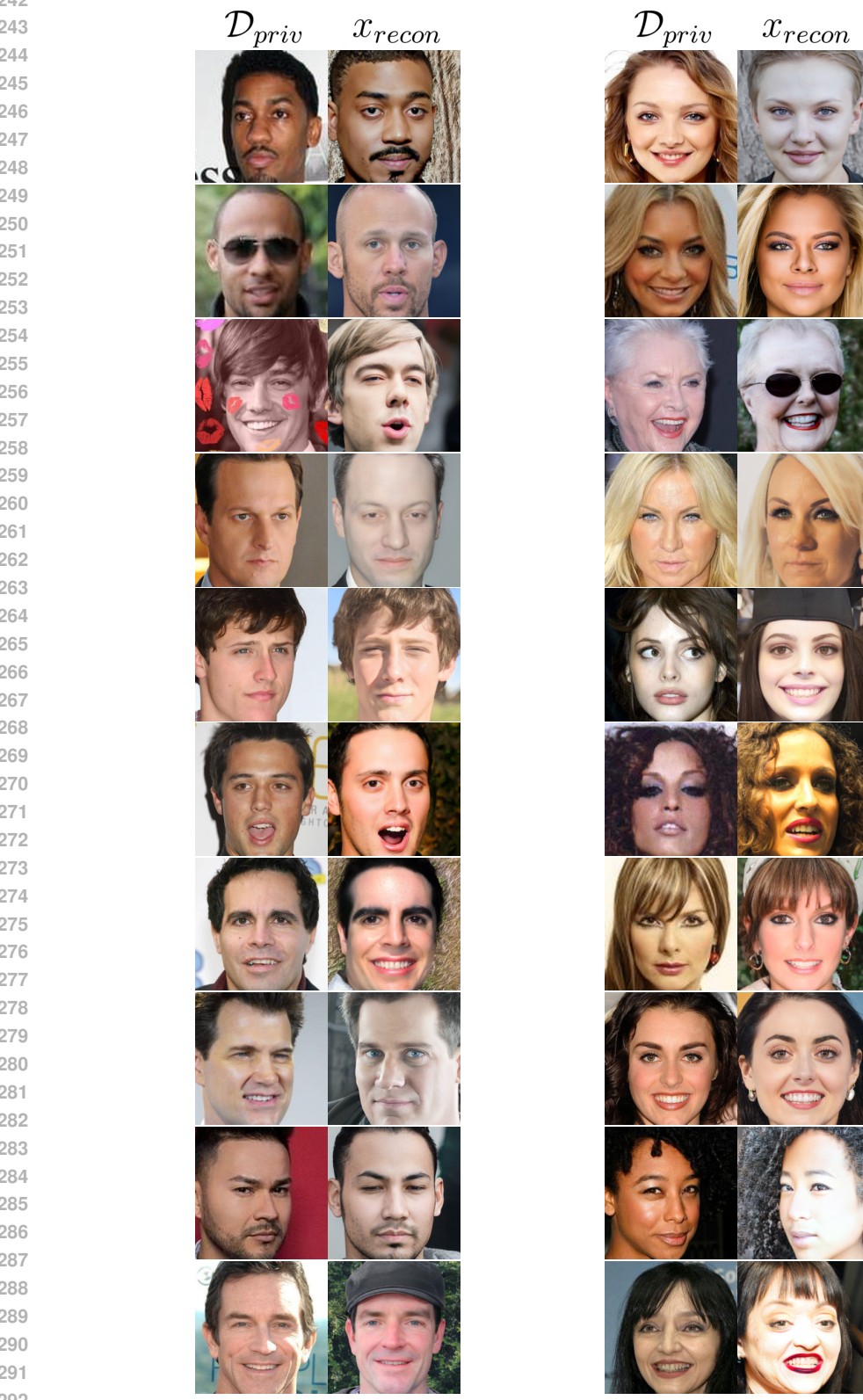

$\mathcal{D}_{priv}$ $\quad$ $x_{recon}$ $\qquad$ $\mathcal{D}_{priv}$ $\quad$ $x_{recon}$

Figure S.5: Qualitative results on CelebA dataset using the SMI-AW and $\mathcal{L}_{LOM}$, $M$ = LLaVA-v1.6-7B. For each pair, the left column shows images from the private training dataset, while the right column presents the reconstructed images corresponding to each individual in the left column.

$$\mathcal{D}_{priv} \qquad x_{recon} \qquad\qquad \mathcal{D}_{priv} \qquad x_{recon}$$

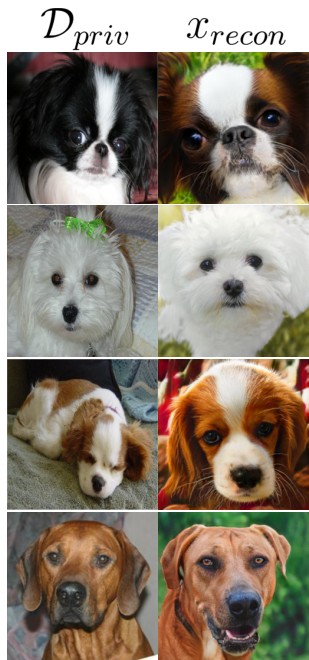 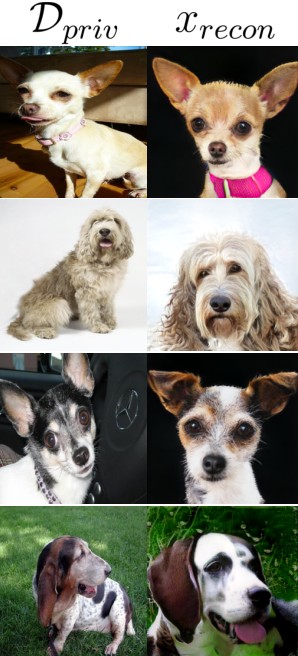

Figure S.6: Qualitative results on the Stanford Dogs dataset using the SMI-AW and $\mathcal{L}_{LOM}$, $M$ = LLaVA-v1.6-7B. For each pair, the left column shows images from the private training dataset, while the right column presents the reconstructed images corresponding to each dog breed in the left column.

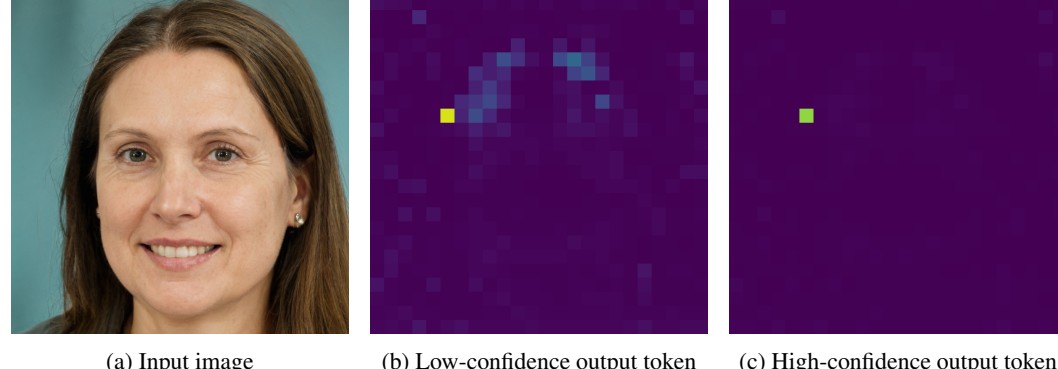

(a) Input image  (b) Low-confidence output token  (c) High-confidence output token

Figure S.7: An illustration of attention maps of low-confidence ($\mathbb{P}(y_i) = 0.1854$) and high-confidence ($\mathbb{P}(y_i) = 0.9999$) output tokens

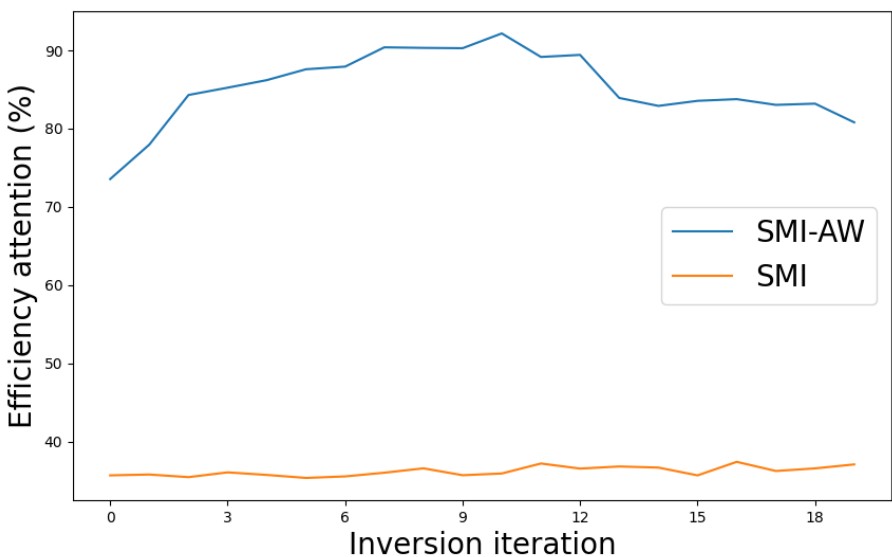

Figure S.8: Percentage of efficient visual attention tokens among all tokens used during inversion for SMI and SMI-AW. With the adaptive weighting, SMI-AW effectively increasing the percentage of efficient visual attention tokens used during inversion. This strategy significantly improves attack accuracy, achieving stronger results across multiple metrics and datasets (Section 4.2).

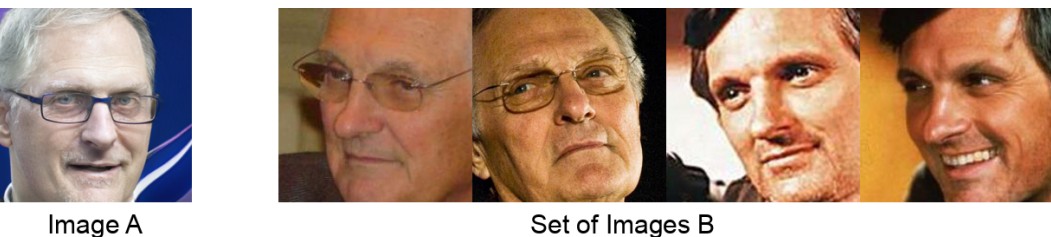

Figure S.9: An example evaluation query in $\mathcal{F}_{MLLM}$ and human evaluation involves determining whether "Image A" depicts the same individual as those in "Image B." "Image A" is a reconstructed image of a target textual answer $y$, while "Image B" contains four real images of the same target textual answer $y$. Gemini or human evaluators respond with "Yes" or "No" to indicate whether "Image A" matches the identity shown in "Image B."

