# OpenReview forum: "Model Inversion Attacks on Vision-Language Models: Do They Leak What They Learn?"
_ICLR.cc/2026/Conference — ICLR 2026 Conference Withdrawn Submission_

### Official Review · Reviewer_mMDv · 2025-10-15

**Soundness:** 2
**Presentation:** 2
**Contribution:** 2
**Rating:** 2
**Confidence:** 5

**Summary:**

This paper proposes the first model inversion attack (MIA) against vision-language models. The paper introduces two token-based MI, including TMI and TMI-C, and two sequence-based MI, including SMI and SMI-AW. Extensive experiments show the effectiveness of the propose method.

**Strengths:**

1. The paper is the first to study MIAs against VLMs.
2. This paper comprehensively considers various possible optimization methods

**Weaknesses:**

1. The assumption that the attacker knows the $t,y$ is unrealistic. In practical scenarios, such detailed knowledge is rarely available to an adversary.
2. The constructed VQA dataset appears too simple. It would be more convincing to evaluate the method under more diverse and natural prompts such as *“Who is the man in the photo?”* or *“What is the person’s name in the image?”*, as well as with more varied answer patterns like *“The man in the photo is named XXX”* or *“XXX is in the image.”*
3. The sequence-based method outperforms the token-based methods, but the paper does not provide sufficient explanation or theoretical justification.
4. This article only arranges and combines various possible optimization methods, but lacks sufficient explanation and theoretical analysis. The contribution therefore feels closer to a technical report rather than a fully developed scientific study.

**Questions:**

1. Why the paper assumes that the attacker knows exact $y$?
2. Could the authors provide experiments or discussions under these more complex and realistic settings, according to W2?
3. Why does the sequence-based method perform better than the token-based ones? Could you provide theoretical justification?
4. In SMI-AW, the paper uses a hard threshold. Have the authors considered a soft-weighting approach, such as those used in focal loss or label smoothing? Are there any theoretical analysis?

---

### Official Review · Reviewer_Mqd2 · 2025-10-30

**Soundness:** 3
**Presentation:** 3
**Contribution:** 3
**Rating:** 4
**Confidence:** 3

**Summary:**

This paper investigates model inversion attacks on Vision-Language Models (VLMs), aiming to reconstruct private training images from multimodal models by exploiting gradients or output probabilities. The authors propose a family of inversion strategies—Token-based (TMI), Convergent Token-based (TMI-C), Sequence-based (SMI), and Sequence-based with Adaptive Weighting (SMI-AW)—to handle the sequential nature of text generation in VLMs. They also design new logit-based loss functions (LMML and LLOM) to improve convergence stability and semantic fidelity. Experiments on multiple VLMs (LLaVA-1.6-7B, Qwen2.5-VL-7B, MiniGPT-v2) and datasets (FaceScrub, CelebA, Stanford Dogs) show that SMI-AW combined with LLOM achieves the most consistent reconstructions across automatic, model-based, and human evaluations. The study demonstrates that VLMs can memorize fine-grained visual details, revealing non-trivial privacy risks even in large open-domain models.

**Strengths:**

1. Novel adaptation of model inversion to multimodal generation.
The paper is among the first to systematically adapt model inversion from classification settings to language-generative VLMs, carefully addressing token-level vs. sequence-level optimization and designing differentiable objectives that align with text decoding processes.

2. Comprehensive empirical study and method variants.
Four inversion strategies and two loss formulations are compared across several open-source VLMs, providing a broad and reproducible experimental landscape. The analysis of token-wise vs. sequence-wise behaviors offers useful insight for future multimodal privacy research.

3. Relevance to privacy and safety auditing of VLMs.
The work highlights realistic risks of information leakage from pretrained vision-language models—an emerging concern as such models are increasingly deployed in consumer and enterprise systems—and thus provides a timely contribution to responsible AI and privacy evaluation research.

**Weaknesses:**

1. The evaluation protocol may overestimate “leakage success,” which the authors should explicitly discuss.
The paper primarily relies on a DNN-based classifier (AttAccD), an MLLM-based judge (AttAccM), and human evaluation to measure attack success. However, prior work shows these automated metrics can suffer from cross-domain bias, prompt sensitivity, and overconfidence, often producing false positives that inflate the reported success rates. The authors should discuss this limitation and consider adding control experiments (e.g., label/answer shuffling, mismatched prompts, adversarial perturbations) or reporting confidence intervals and statistical significance to better quantify evaluation reliability.

2. Experimental scope is narrow; external validity should be discussed.
All experiments are conducted on public datasets such as FaceScrub, CelebA, and Stanford Dogs under white-box assumptions. While this setup demonstrates the upper bound of potential privacy risk, it does not represent realistic sensitive or black-box deployment scenarios (e.g., medical or enterprise models). The authors should discuss the applicability boundaries of their attack in more restricted access settings (logit-only, API-based) and justify their dataset choices. It would also be useful to mention plans to test on non-public or cross-domain tasks to assess real-world transferability.

3. Generator dependence is under-analyzed; its impact on robustness should be discussed.
The attack is entirely optimized in the StyleGAN2 latent space, yet prior studies show that different priors—especially diffusion-based ones—can drastically affect inversion quality and bias the generated images. Without cross-prior or multi-initialization consistency checks, it is difficult to disentangle true VLM memorization from generator artifacts. The authors should explicitly discuss this dependency and its implications for the paper’s conclusions, and ideally explore diffusion-based priors or sensitivity analyses in future work to substantiate that the recovered signals indeed stem from the VLM’s internal memory rather than the generator’s bias.

**Questions:**

+ While the paper demonstrates that model inversion can reconstruct semantically consistent images from VLMs, it remains unclear what the practical implications of these attacks are in real-world settings. Specifically, could the authors clarify how to distinguish between meaningful privacy leakage and general semantic reconstruction (e.g., when the recovered image reflects public knowledge like a celebrity’s appearance rather than a memorized training sample)? In other words, what does a “successful inversion” truly imply for the privacy or safety of VLM deployments, and how should practitioners interpret these results when assessing actual risk?

+ The paper “Revisiting Model Inversion Evaluation: From Misleading Standards to Reliable Privacy Assessment” argues that many prior MI works—including multimodal variants—overestimate privacy leakage due to flawed evaluation metrics and lack of proper baselines or controls. Could the authors clarify how yours differs from or improves upon the critiques raised in that work? In particular, how does this paper ensure that its reported inversion success rates reflect true memorization rather than artifacts of evaluation bias (e.g., classifier dependence, prompt bias, or semantic overlap)?

---

### Official Review · Reviewer_n3c6 · 2025-10-31

**Soundness:** 2
**Presentation:** 4
**Contribution:** 3
**Rating:** 4
**Confidence:** 4

**Summary:**

This paper investigates Model Inversion (MI) attacks against Vision-Language Models (VLMs) to assess their potential privacy risks. Given white-box access to a VLM, a public dataset, input prompts, and target textual outputs, the attack aims to reconstruct corresponding private training images.

To tailored VLMs' token-based generation, the authors propose four attack strategies: two token-based and two sequence-based. The Sequence-based Model Inversion with Adaptive Token Weighting (SMI-AW), which prioritizes low-confidence tokens, achieves the best performance across various models and datasets. Experiments report up to 75% attack success rate under human evaluation, demonstrating that VLMs leak identifiable visual information from their training data.

**Strengths:**

1. **Revealing VLMs' privacy risk**: The paper is the first exploration of MI attacks on VLMs. It fills an important research gap, exposing potential privacy vulnerabilities in VLMs.
2. **Novel attack frameworks**: The proposed token- and sequence-based inversion frameworks adapt conventional MI techniques to VLMs' sequential token generation process. The introduction of adaptive token weighting (SMI-AW) is well-motivated and empirically effective.
3. **Comprehensive and clear presentation**: The algorithms, loss functions, and evaluation metrics are clearly described, making this work easy to follow.

**Weaknesses:**

1. **Strong attacker assumptions**: The threat model assumes access to the white-box model, a public dataset, target input prompts and ground-truth outputs, which limits real-world applicability. It would strengthen the work to explore (or at least discuss) more realistic cases, such as:
   1. Black-box settings
   2. Attacks without auxiliary public data
   3. Attacks without access to textual ground-truth samples
2. **Simplified task formulation**: The VQA tasks in experiments are effectively classification problems  (e.g., prompts like "Who is the person in the image?" with identity-name answers). This setup may overstate inversion success. It remains unclear whether similar attacks would succeed on richer, more diverse multimodal datasets.
3. **Potential bias in human evaluation**: I appreciate the human evaluation. However, in my own opinion, the binary question ("Does A depict the same individual as B?") can yield ambiguous responses. Including baseline comparisons (e.g., public samples of similar but different identities or attacks on pre-trained models) would better contextualize the evaluation results.
4. **Limited analysis from the victim's perspective**: The ablation studies focus mainly on attack-side configurations. Examining how factors such as fine-tuning data size, model capacity, or training duration affect vulnerability would deepen understanding of privacy risks.
5. **No defense exploration**: While this is primarily an attack paper, evaluating SMI-AW under basic defenses (e.g., differential privacy, weight clipping) or discussing mitigation strategies would enhance its practical impact.

**Questions:**

1. **Justification of the re-weighting strategy**: Supp C links adaptive weights to visual attention efficiency. Have the authors tried directly re-weighting tokens based on visual attention scores instead of confidence? Would this further improve inversion quality?
2. **Data separation between pre-training and fine-tuning dataset**: Could the author confirm the fine-tuning datasets are disjoint from the VLMs' pre-training data? If not, overlapping identities (e.g., celebrities or dog breeds) might confound claims that the attack reveals fine-tuning data rather than pre-training artifacts.
3. **Pre-training vs. fine-tuning leakage**: Do the authors observe any difference in privacy risk between the pre-trained and fine-tuned stages? For instance, does one expose more information than the other?
4. **Architectural sensitivity**: Table 1 and 3 show that attack success varies across LLaVA-v1.6, Qwen2.5VL, and MiniGPT-v2. Could the authors elaborate on what architectural factors might influence inversion vulnerability?

---

### Official Review · Reviewer_Csri · 2025-10-31

**Soundness:** 2
**Presentation:** 2
**Contribution:** 2
**Rating:** 2
**Confidence:** 4

**Summary:**

This paper presents the first systematic study of model inversion (MI) attacks on vision-language models (VLMs), investigating whether multimodal models leak private visual data from their training sets. The authors identify that existing MI attacks mainly target unimodal DNNs and are not directly applicable to VLMs due to token-based text generation and multimodal training.

To address this, the paper introduces four new attack strategies tailored for VLMs. Among these, SMI-AW dynamically reweights token-level losses according to prediction confidence, emphasizing low-confidence tokens that provide stronger feedback for reconstruction. Experiments are conducted on three VLMs (LLaVA-v1.6-7B, Qwen2.5-VL-7B, MiniGPT-v2) and three datasets (VQA-FaceScrub, VQA-CelebA, VQA-Stanford Dogs).

**Strengths:**

1: The paper explores a relatively under-studied topic, i.e., privacy leakage through model inversion (MI) attacks in multimodal VLMs. Thereby addressing a notable gap between existing unimodal and multimodal privacy research.

2: The paper introduces four distinct strategies for performing MI attacks.

**Weaknesses:**

1: According to Algorithm 1 and Algorithm 2, the two methods appear to perform essentially the same procedure, differing only in the order of the nested loops. Please clarify whether there is any substantive difference in functionality or outcome between the two algorithms.

2: The overall presentation of the paper could be made more concise and clearer.

**Questions:**

1: Please provide a clearer explanation of Weakness 1.

2: Among the four strategies introduced in the paper, which one is actually used during training? The description is ambiguous. It is unclear whether all are applied jointly, sequentially, or if only a subset is used.

3: Please clarify which loss function is employed. Is it a cross-entropy–based loss or a logit-based objective?

---

### Official Review · Reviewer_ojj7 · 2025-11-01

**Soundness:** 3
**Presentation:** 3
**Contribution:** 3
**Rating:** 4
**Confidence:** 5

**Summary:**

This paper presents the first systematic study of Model Inversion (MI) attacks on Vision-Language Models (VLMs), addressing a critical research gap where prior MI work has focused exclusively on unimodal DNNs. The author proposes two token-based methods (TMI, TMI-C) and two sequence-based methods (SMI, SMI-AW). Experimental results have edemonstrated the effectiveness of the proposed methods.

**Strengths:**

- Contribution: It is the first study to systematically investigate MI attacks on VLMs.
- Methodological Innovation: The four tailored MI strategies (especially SMI-AW) provide a new framework for attacking token-generating multimodal models.
- Writting: This paper is well-structured and easy to follow.

**Weaknesses:**

- The study focuses exclusively on facial images (FaceScrub/CelebA) and dog breeds (Stanford Dogs). It does not test more diverse or sensitive domains (e.g., medical images, satellite data), which are critical for assessing current VLMs’ privacy risks in high-stakes applications.
- Only three VLMs are evaluated. Testing additional architectures (especially large vision language models with large parameter count) would strengthen claims about VLMs’ general vulnerability。
- While the work calls for defenses, it does not explore even preliminary mitigation strategies (e.g., regularization, token-level noise injection).  More evaluation on defenses are expected.

**Questions:**

See Weaknesses.

---

### Note · Authors · 2025-11-14

I have read and agree with the venue's withdrawal policy on behalf of myself and my co-authors.